# Numerical Simulation of Bottom-Blowing Stirring in Different Smelting Stages of Electric Arc Furnace Steelmaking

**Hang Hu, Lingzhi Yang \*, Yufeng Guo, Feng Chen, Shuai Wang, Fuqiang Zheng and Bo Li**

School of Minerals Processing and Bioengineering, Central South University, Changsha 410083, China; csu-huhang@csu.edu.cn (H.H.); yfguo@csu.edu.cn (Y.G.); csuchenf@csu.edu.cn (F.C.); wang_shuai@csu.edu.cn (S.W.); f.q.zheng@csu.edu.cn (F.Z.); lee0109@csu.edu.cn (B.L.)
* Correspondence: yanglingzhi@csu.edu.cn

**Abstract:** Electric arc furnace (EAF) steel bottom-blowing can effectively improve the temperature and composition uniformity of the molten pool during smelting process. To explore the effect of molten-steel characteristics on bottom-blowing at various stages of smelting, we divided the smelting process of the EAF into four stages: the melting stage, the early decarburization stage, the intermediate smelting stage, and the ending smelting stage. The numerical simulation software ANSYS Fluent 18.2 was used to simulate the velocity field of molten steel under the condition of bottom-blowing stirring in different stages in EAF steelmaking process. The properties of bottom-blowing and the kinetic conditions of the steel-slag at this interface were investigated. Our results showed that at a bottom-blowing gas flow rate of 100 L/min, the average flow rates of the four stages were v1 = 0.0081 m/s, v2 = 0.0069 m/s, v3 = 0.0063 m/s, and v4 = 0.0053 m/s. The physical model verification confirmed the results, that is, the viscosity of molten steel decreased as the smelting progressed, and the flow velocity of the molten steel caused by the agitation of bottom-blowing also decreased, the effect of bottom-blowing decreased. Based on these results, a theoretical basis was provided for the development of the bottom-blowing process.

**Keywords:** electric arc furnace steelmaking; bottom-stirring; different smelting time; molten steel flow; numerical simulation



## 1. Introduction

Electric arc furnace (EAF) steelmaking, as the core of the short-process steelmaking method, has the characteristics of having a short process, low energy consumption, diversified charge, and diversified product structure [1,2]. The flow characteristics of molten steel in electric arc furnace (EAF) steelmaking remarkably influence heat transfer, momentum transfer, mass transfer, and chemical reactions among different phases in the smelting process [3–5]. The appropriate flow characteristics depend on the temperature and chemical composition of molten steel and determine the success of the smelting process. Disadvantages such as small free space, weak stirring ability, and uneven distribution of composition and temperature in the arc furnace can be improved by applying bottom-blowing technology in molten pool stirring. Blowing Ar, $N_2$, $CO_2$, and other gases into the molten pool through the gas-supply component arranged at the bottom of the arc furnace can strengthen mixing in the molten pool, increase the reaction contact area, promote the steel-slag reaction, and accelerate metallurgical speed. Thus, it is of great significance to improve the quality of molten steel and metal yield [6–10].

The flow characteristics of molten steel under bottom-blowing technology in EAF smelting have been extensively studied. The disadvantages of low mass and heat transfer rates caused by poor stirring in the molten pool during EAF steelmaking have been improved after the adoption of bottom-blowing technology [11]. Ramírez [12] established a mathematical model to describe the flow in the molten pool of a direct current (DC) electric arc furnace and found that the maximum temperature of liquid steel rose from 1966 to

1999 K after the application of bottom-blowing, and the maximum flow velocity of liquid steel rose from 0.8 to 4 m/s, which significantly improved the flow of the molten pool and smelting effect. Detailed analyses on the effect of the bottom-blowing system, through numerical simulation, experimental research, and industrial applications, were conducted by the Zhu group at the University of Science and Technology Beijing [13–17]. Dong [14] studied the application effect of bottom-blowing gas at a flow rate of $0.3-0.9$ N m$^3$/h in 70 t EAF and revealed decarburization speed increased from 0.04 to 0.10 %/min, and lime consumption reduced by 10.4 kg/t. The bath blended time is only $1.7-2.5$ min. Wei [15] studied the velocity distribution in the molten bath at different bottom-blowing gas flow rates and revealed the velocity of molten steel increased when the bottom-blowing gas rates increased, the content of phosphorus in the molten steel was decreased by 0.005 wt.%, the contents of FeO and T. Fe in endpoint slag were, respectively, reduced by 4.1 and 4.7 wt.%, the dephosphorization and decarburization rate were, respectively, increased by 12.1 and 11.8%, and the endpoint carbon–oxygen equilibrium of the molten steel was improved by 0.0024. Liu [16] researched stirring effects of six kinds of bottom-blowing arrangements on the molten bath in a 75 t EAF, the results showed increasing flow rate, weakening impeding force of sidewall, and improving stirring effect on the molten bath in eccentric bottom-tapping (EBT) region would decrease mixing time and improved stirring ability. Ma [17] studied the flow field with bottom-blowing in a 70 t EAF, and the numerical simulation studies showed that the turbulent kinetic energy of the EAF molten pool was increased by 87.9% and metal velocity was increased by 98%. The dead area decreased by 79%. When using the combined blowing technology, it can significantly improve the stirring intensity and the mixing effect of liquid steel, reducing smelting cost.

The above-mentioned research results preliminarily show that bottom-blowing stirring can enhance the flow characteristics of molten steel and promote smooth smelting. Because EAF smelting is performed at high temperatures, the dynamic characteristics of the molten pool in different smelting stages have not been well understood, and studies concerned the molten pool state in a single stage. In the smelting process, the bottom- blowing flow rate is usually constant, while the composition, temperature, and physical characteristics of molten steel change. The influences of molten steel with different physical characteristics affected the molten pool differently under the same bottom-blowing conditions. Therefore, at a certain stage of smelting, it is easy to cause insufficient stirring intensity due to insufficient bottom-blowing gas or the scouring of the furnace lining and waste of bottom-blowing gas.

Thus, determining the main flow properties of molten steel in different stages and studying the effect of bottom-blowing stirring on molten steel can help understand the bottom-blowing effect in different stages of the EAF smelting process, which is conducive to improving the utilization rate of the bottom-blowing gas and forming a better smelting effect. In this paper, numerical simulation and model verification were used to study the influence of molten steel characteristics in different smelting periods on the flow rate of molten steel under the same bottom-blowing conditions.

## 2. Flow Characteristics of Molten Steel in Different Smelting Periods of EAF

Modern electric arc furnace smelting retains the main melting, heating, and necessary refining processes. Dephosphorization and partial decarburization are advanced to the early stage of smelting as far as possible. In the middle and late stages of smelting, only the carbon content and temperature of steel and slag are controlled, and the tap-to-tap time is shortened. In a short period, the steel scrap is melted, and the temperature of the molten steel is adjusted to meet the requirements of tapping. Due to the current use of eccentric bottom-tapping (EBT) tapping and retained steel operation, the molten pool already exists when the steel scrap melts. Combined with enhanced oxygen jet technology and bottom-blowing stirring technology, the EAF offers favorable conditions for the metallurgical reaction.

### 2.1. Factors Influencing Liquid Steel Flow Characteristics

Decarburization, dephosphorization, and metal oxidation reactions occur in EAF. The related operation system and the corresponding temperature and composition changes of molten steel and slag in the steelmaking process change the physical characteristics of molten steel and, consequently, the gas stirring effect. Density and viscosity are the main physical characteristics of molten steel, and its main influencing factors include chemical composition and temperature.

Density is a basic variable to explain the behavior and properties of liquid metals, and it is a basic parameter to reflect and analyze the melt structure. Systematic studies on the density of various liquid metals have been performed [18–23]. Viscosity is the momentum of fluid molecules moving from the current liquid layer to another liquid layer perpendicular to the streamline direction. Viscosity is one of the important physical properties of liquid metals, determining their hydrodynamic characteristics [24]. It also has a great influence on heat, momentum, and mass transfer and chemical reaction between phases in EAF steelmaking process. The Roscoe formula modified by Iida has high reliability and accuracy in calculating the viscosities of liquid metals, such as molten steel and liquid iron [25–27].

Studies on the density and viscosity of liquid iron and molten steel were mainly based on experimental design and theoretical analysis, to determine a relative value, without coupling the influences of multi-fluid and multi-phase interactions in the steelmaking process. Stirring methods commonly used in EAF steelmaking process, such as oxygen jet [28], bottom-blowing stirring, and electromagnetic stirring [29], can strengthen the momentum transfer and heat and mass exchange of molten steel in the furnace by inputting material and energy and improve the fluidity of molten pool to a certain extent. However, the molten pool flow is ultimately determined by the flow characteristics of molten steel. Excessive external input is inefficient when the internal molten steel viscosity only slightly changes, causing material and energy consumption. Therefore, analyzing the influencing factors of the liquid steel viscosity in EAF steelmaking requires understanding the effect of bottom-blowing stirring to improve smelting efficiency.

The viscosity of molten steel is affected by the temperature and chemical elements content of molten steel, especially the temperature. With the temperature range from melting point to 1923 K, the relationship between the viscosity and temperature [24] is shown as follows:

$$lg\eta = \frac{1951}{T} - 3.327 \tag{1}$$

where $\eta$ is the viscosity of molten steel, Pa·s; $T$ is the temperature of the molten steel, K; with the temperature increase, the viscosity of molten steel decreases. When the temperature increases, the momentum of atoms in molten steel jumping into the adjacent liquid layer increases, providing enough energy for particles to move so that the number of particles with the viscous flow activation energy increases, and the viscosity of the melt decreases.

### 2.2. Feature Point Selection in Different Smelting Periods

EAF steelmaking is accompanied by many physical changes and chemical reactions. In the initial stage of smelting, the scrap melts gradually and then forms the molten pool, and silicon and manganese in the molten steel are first oxidized and then floated to the molten steel surface or suspended in the molten steel. The dephosphorization agent, CaO, carries oxygen or argon and disperses it in the molten steel in the form of powder particles for the dephosphorization reaction. In this process, the power of electrode heating is larger, heating the molten pool and increasing the temperature of it with the maximum power, which is conducive to the dephosphorization reaction. In the middle of smelting, the molten pool begins to skim slag, slowing down the heating rate, increasing the alkalinity of steel slag, and increasing the oxygen supply rate, to improve dephosphorization conditions. After the dephosphorization reaction stops, arc heating is stopped, and the supply of oxygen is reduced. At the same time, many phosphorus-containing slags are stripped to

prevent phosphorus recovery. During the dephosphorization reaction, the decarburization reaction also accompanies it, and the iron element in the molten steel is oxidized. At a later stage of smelting, oxygen is supplied at a medium rate, and the electric arc furnace is used for only high-power heating to facilitate the decarburization reaction and adjust the temperature. At the end of smelting, the oxygen supply and heating are stopped, the composition and temperature of molten steel are stabilized, and the steel is prepared for reduction refining.

According to the various characteristics of elements and temperature in the EAF steelmaking process, four liquid steel components and temperatures were selected in this work as the characteristic points of the four smelting stages. The theoretical values of the liquid steel density and viscosity were calculated based on Equation (1) and Reference [30]. Before the EAF steelmaking process, the chemical composition of hot metal and scrap would be tested and analyzed. Then, the initial carbon and silicon contents of the bath could be obtained by dividing the total mass of carbon and silicon by the total weight of molten steel, respectively. The specific parameters are shown in Table 1.

**Table 1.** Relevant parameters of characteristic points in different smelting periods.

| Physical Properties | Temperature (°C) | Carbon Content (%) | Silicon Content (%) | Viscosity (Pa·s) | Density (g/cm$^3$) |
|---|---|---|---|---|---|
| Point 1 (Melting stage) | 1140 | 2.581 | 0.3165 | 0.0113 | 6.900 |
| Point 2 (Early decarburization stage) | 1400 | 2.3259 | 0.0593 | 0.0069 | 6.662 |
| Point 3 (intermediate smelting stage) | 1500 | 1.132 | 0 | 0.0059 | 7.003 |
| Point 4 (Ending smelting stage) | 1650 | 0.1035 | 0 | 0.0049 | 6.912 |

## 3. Numerical Simulation of Bottom-Blowing

### 3.1. Governing Equation

The molten steel, slag, and argon need to satisfy not only the mass, energy, and momentum conservation equations but also the control equation of the finite element model and turbulence control equation. To study the effect of bottom-blowing gas agitation on the molten steel flow rate in the EAF steelmaking process, a fluid volume function (VOF) model calculation was introduced during the establishment of the CFD model. In the simulation process, different fluid equation components share a set of momentum equations. The effective density $\rho_e$ of the argon–slag–steel liquid (three-phase) system explored in this study was derived from the following equation:

$$\rho_e = \alpha_{Ar}\rho_{Ar} + \alpha_{Sl}\rho_{Sl} + \alpha_{St}\rho_{St} \tag{2}$$

where $\alpha_{Ar}$, $\alpha_{Sl}$, and $\alpha_{St}$ (%), $\rho_{Ar}$, $\rho_{Sl}$, and $\rho_{St}$ (kg/m$^3$) are the volume fractions and the densities of argon, slag, and molten steel, respectively.

The energy equation in the calculation domain is shared by all phases, and the expression is as follows:

$$\frac{\partial(\rho E)}{\partial t} + \nabla\cdot(u(\rho E + p)) = \nabla\cdot\left(k_{eff}\nabla T\right) + S_h \tag{3}$$

where $\rho$ is the density of the gas, kg/m$^3$; $E$ is the element energy, J; $t$ is the time, **s**; $p$ is the static pressure of the fluid, MPa; $T$ is the temperature, K; $k_{eff}$ is the effective thermal conductivity, W/(m·K); $\rho_e$ and $k_{eff}$ are shared with all phases; and the source term $S_h$ is provided by radiant heat transfer and other volumetric heat sources.

In the VOF model, the energy $E$ can be obtained by the mass weighted average method which is described by the following formula.

$$E = \sum_{i=1}^{n} \alpha_i \rho_i E_i \bigg/ \sum_{i=1}^{n} \alpha_i \rho_i \tag{4}$$

where $\alpha_i$ (%) and $\rho_i$ (kg/m$^3$) are the volume fraction and density of phase $i$, respectively; and $E_i$ is based on the specific heat and shared temperature for each phase. The standard $k - \varepsilon$ turbulence model was used in this study. The turbulent flow energy $k$ (m$^2$/s$^{-2}$) and the dissipation rate $\varepsilon$ (m$^2$/s$^3$) were determined by the following transfer equations, respectively.

$$\frac{\partial(\rho k)}{\partial t} + \frac{\partial(\rho k v_i)}{\partial x_i} = \frac{\partial}{\partial x_j}\left[\left(\mu + \frac{\mu_t}{\sigma_k}\right) \cdot \frac{\partial k}{\partial x_i}\right] + G_k + G_b - \rho\varepsilon - Y_M + S_k \tag{5}$$

$$\frac{\partial(\rho \varepsilon)}{\partial t} + \frac{\partial(\rho \varepsilon v_i)}{\partial x_i} = \frac{\partial}{\partial x_j}\left[\left(\mu + \frac{\mu_t}{\sigma_\varepsilon}\right) \cdot \frac{\partial \varepsilon}{\partial x_i}\right] + C_{1\varepsilon}\frac{\varepsilon}{k}(G_k + C_{3\varepsilon}G_b) + C_{2\varepsilon}\rho\frac{\varepsilon^2}{k} + S_\varepsilon \tag{6}$$

In these equations, $\mu$ represents the dynamic viscosity, Pa·s; and $x_i$ and $x_j$ represent the coordinates along the coordinate axis $i$ and $j$ directions, respectively;

$G_k$ and $G_b$ are turbulent flow energies generated by the average fluid velocity and buoyancy, J, respectively; $Y_M$ is the turbulent dissipation rate generated by a compressible turbulent pulsation; $S_k$ and $S_\varepsilon$ are custom source terms. The expression for calculating the turbulent viscosity $\mu_t$ (Pa·s) using $k$ and $\varepsilon$ is as follows:

$$\mu_t = \rho C_\mu \frac{\varepsilon^2}{k} \tag{7}$$

where $C_{1\varepsilon}$, $C_{2\varepsilon}$, $C_{3\varepsilon}$, $\sigma_k$, $\sigma_\varepsilon$, and $C_\mu$ are the constant terms of the $k - \varepsilon$ model, and their values were provided by Launder, as 1.44, 1.92, 0.8, 1.0, 0.9, and 0.009, respectively [31]. The continuity equation is as follows:

$$\frac{1}{\rho_i}\left[\frac{\partial}{\partial t}(\alpha_i \rho_i) + \nabla \cdot (\alpha_i \rho_i v_i)\right] = S_{\alpha_i} + \sum_{i=1}^{n}(m_{ji} - m_{ij}) \tag{8}$$

where $v_i$ is the velocity component in direction $i$, m/s; $m_{ij}$ is the mass transmitted from phase $i$ to phase $j$, kg; $m_{ji}$ is the mass transmitted from phase $j$ to phase $i$, kg; and $S_{\alpha_i}$ is a custom source item. The momentum equation is as follows:

$$\frac{\partial}{\partial t}\left(\rho\vec{v}\right) + \nabla \cdot \left(\rho\vec{v}\vec{v}\right) = -\nabla p + \nabla \cdot \left[\mu\left(\nabla\vec{v} + \nabla\vec{v}^T\right)\right] + \rho\vec{g} + F \tag{9}$$

where $\vec{v}$ is the instantaneous velocity of the fluid, m/s; $p$ is the static pressure, MPa; $g$ is the acceleration of gravity, m/s$^2$; and $F$ is the other force that the control body receives, N.

*3.2. Grid Model*

A 100 t industrial top-charge EAF in a steel plant was selected to study the physical and chemical properties of molten steel in the EAF steelmaking process. To ensure the reliability of the results, a grid calculation model with a ratio of 1:1 was established using numerical simulation software. Based on the previous research results, the optimized bottom-blowing layout was set in this study. The geometry and bottom-blowing layout of the 100 t EAF are shown in Figure 1.

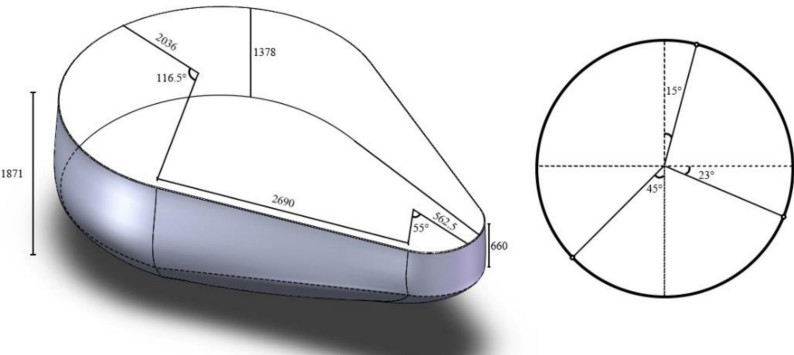

**Figure 1.** Geometric parameters and bottom blowhole arrangement of the Electric arc furnace (EAF).

In the geometric model, the diameter of the reference circle on the top surface of the eccentric area was 1374 mm, the diameter of the reference circle on the top surface of the main body of the EAF was 4072 mm, the diameter of the bottom circle of the main body of the EAF was 2122 mm, and the furnace length of the EAF was 6355 mm. The length of the hypotenuse was 2690 mm, and the actual fan angles at the top of the EAF model were 233° and 110°, respectively. The bottom-blowing hole has a hydraulic radius of 10 mm and is arranged on a circumference of 1001.1 mm from the center; the hydraulic radius of the molten pool was 5481 mm, the depth was 1650 mm, the internal molten steel depth was 1430 mm, and the thickness of the slag phase was 165 mm. The number of meshes in the mesh model established based on the geometric model was 55,924, and the number of nodes was 540,395 [31] (Figure 2).

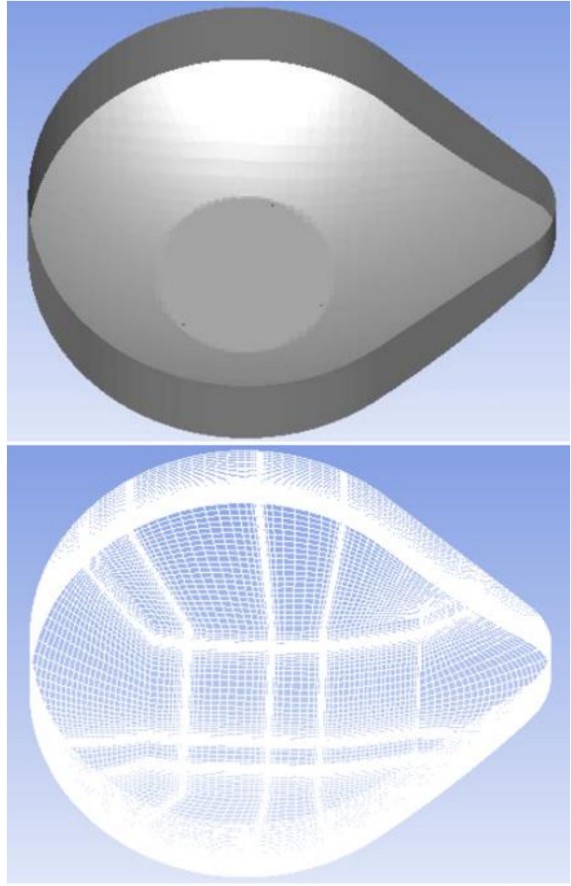

**Figure 2.** Simulation and grid diagram of the EAF.

### 3.3. Calculation Hypothesis

(1) The slag, molten steel, and gas phases formed a transient non-isothermal three-phase flow fluid, and the three phases were immiscible.;

(2) The gas phase was regarded as a compressible Newtonian fluid, and the slag and steel liquid phases were regarded as incompressible Newtonian fluids;

(3) All the walls during the simulation were non-smooth wall models, and the average velocity of the fluid close to the wall was simulated by the standard wall model;

(4) The simulation process did not consider the chemical reactions occurring between the components.

### 3.4. Calculation Settings

The materials involved in this work include argon, molten steel, and slag. Table 1 involves the related parameters of molten steel, and we set the viscosity of the steel slag to 0.35 Pa·s. The specific heat of argon gas does not change significantly in the temperature range of 298-2000 K; therefore, the values in the built-in database were used. The material properties of steel slag and argon are shown in Table 2.

**Table 2.** Parameter settings of materials in the fluid domain.

| Materials | Slag | Argon |
|---|---|---|
| Density (kg/m$^3$) | 3000 | Ideal-gas |
| Specific Heat (J/(kg·K)) | 1200 | 520.64 |
| Thermal Conductivity (W/m·K) | 1.2 | 0.0158 |
| Viscosity (kg/m·s) | 0.35 [15,31] | $2.125 \times 10^{-5}$ (298.15 K) |
| Molar Mass (kg/mol) | 31.996 | 39.948 |

In the simulation calculation, three mass flow inlet boundaries and a pressure outlet boundary were used to simulate the bottom-blowing flow (100 L/min) of the three bottom-blowing holes and the gas flow and recirculation at the top of the EAF, respectively. The other walls were adiabatic and non-smooth, which were used to simulate the lining and wall parts of the EAF; in addition, a fluid condition was set to simulate the distribution of the molten steel, slag, and argon in the EAF.

To couple pressure and velocity, the most widely used SIMPLE algorithm was adopted. The calculation area was divided into structured grids, and the gradient was calculated based on the least square method of the smallest unit. The pressure calculation was performed using the PRESTO algorithm. The momentum, turbulent kinetic energy, and turbulent dissipation rate and energy were solved by the second-order upwind scheme.

## 4. Results and Discussion

### 4.1. Numerical Simulation Results

The numerical results of each step were calculated, and the ANSYS Fluent 18.2 (version: 18.2, ANSYS lnc., Canonsburg, PA, USA) [32] statistical calculation was outputted in the form of a file. Each step corresponds to the average speed of molten steel. The statistical data greatly fluctuated at the beginning of the calculation, and the data tended to be flat in the later period; thus, the second half of the data that tended to be flat after 200,000 steps was selected as the basis. The average flow rate of this part represents the average flow rate of molten steel after its smooth blowing [31].

Each smelting stage corresponded to the state of the molten pool after the calculation was completed. Since the velocity cloud at the bottom of the arc furnace fluctuated greatly, only the average value of the flow rate of the molten steel in the upper part was used to represent the average flow rate of the molten steel in the entire arc furnace. The molten steel was divided into six planes. At each stage, the velocity clouds of 0.1, 0.2, 0.3, 0.4, 0.5, and 0.6 m were taken from the slag–steel contact surface, as shown in Figure 3. The calculation of the average flow rate of the molten steel at the end of the simulation requires statistical

calculation. A single cloud image was imported into Photoshop 2018, and the color range function was used with the default setting to count the area of the color distribution corresponding to each speed. The average speed was calculated with the area as the weight, which was the average flow rate of the molten steel in that section, and the average value of the corresponding flow velocity of the six planes was calculated as the average value of the molten steel (Table 3).

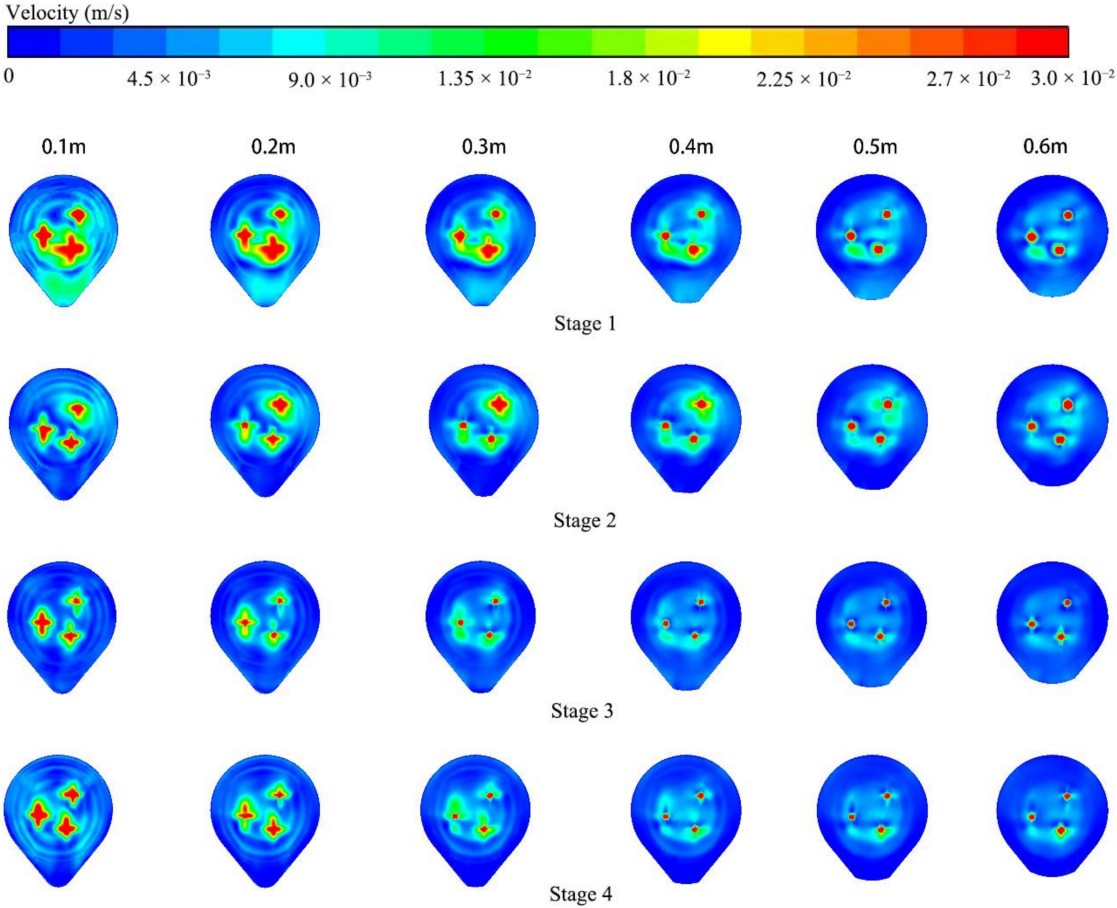

**Figure 3.** Speed cloud map of each interface in different smelting stages.

**Table 3.** Average flow velocity of molten steel at different smelting stages.

| Number of Smelting Stages | Average Flow Velocity of the Liquid Surface at the Corresponding Depth from the Slag–Steel Interface ($\times 10^{-3}$ m/s) | | | | | | Average value ($\times 10^{-3}$ m/s) |
|---|---|---|---|---|---|---|---|
| | 0.1 m | 0.2 m | 0.3 m | 0.4 m | 0.5 m | 0.6 m | |
| 1 | 10.970 | 9.422 | 8.448 | 7.550 | 6.913 | 6.499 | 8.107 |
| 2 | 7.517 | 7.392 | 7.277 | 6.815 | 6.554 | 6.306 | 6.941 |
| 3 | 7.797 | 7.078 | 6.452 | 5.915 | 5.659 | 5.367 | 6.266 |
| 4 | 7.018 | 5.883 | 5.383 | 4.961 | 4.799 | 4.518 | 5.336 |

When the bottom-blowing gas flow rate was 100 L/min, the average flow rates of the four stages were $v_1 = 0.0081$ m/s, $v_2 = 0.0069$ m/s, $v_3 = 0.0063$ m/s, and $v_4 = 0.0053$ m/s. With the progress of smelting, the flow rate of the stirring intensity of the molten steel under the same bottom-blowing conditions gradually decreased.

The average flow rates of the molten steel at different stages and distances from the steel-slag contact surface at different stages are shown in Figure 4. Under the condition that the bottom-blowing gas flow rate remained unchanged, the average molten steel flow rate decreased significantly as the EAF smelting process proceeded. Simultaneously, the

average flow velocity of the molten steel decreased with the distance farther away from the contact surface of the slag–steel.

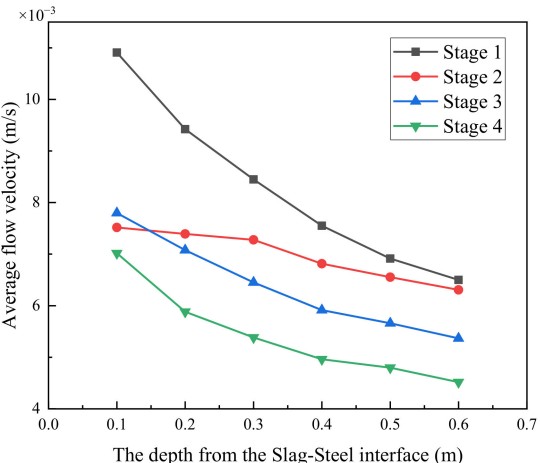

**Figure 4.** Speed distribution at each interface.

According to the simulation data, the viscosity of the molten steel and the average speed were plotted, as shown in Figure 5. At the same time, the curve was linearly fitted to obtain the following equation of molten steel flow velocity and viscosity:

$$v = -85048.18\eta^2 + 1808.75\eta - 1.473 \tag{10}$$

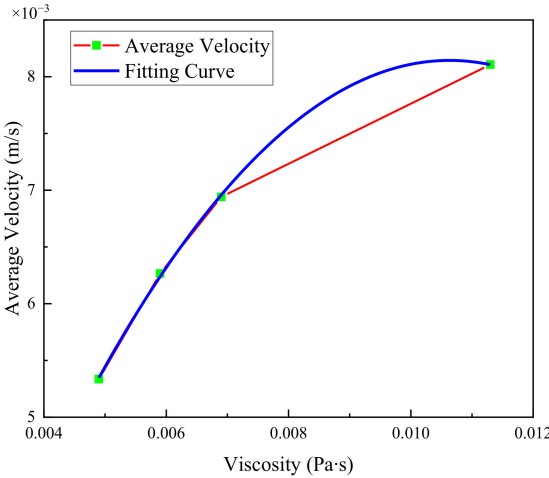

**Figure 5.** Average velocity versus molten steel viscosity diagram and fitting curve.

The sum of squared residuals of the fitted curve was 0.00122, and the range of the squared range was 0.9997, meeting the statistical fitting error requirements. In the formula, $v$ is the flow velocity of the molten steel, $10^{-3}$ m/s; and $\eta$ is the viscosity of the molten steel, Pa·s.

Under the same bottom-blowing gas flow rate, when the viscosity of the molten steel was high in the early stage of smelting, the acceleration effect of the bottom-blowing stirring on the molten steel was better, and the molten steel flow rate was faster. With the progress of smelting, the viscosity of the molten steel decreased, the acceleration effect of the bottom-blowing stirring on the molten steel worsened, and the flow rate of the molten steel decreased.

*4.2. Physical Model Verification*

The numerical simulation results showed that the ongoing smelting process made the molten steel more difficult to be stirred. However, artificial experience and industrial production practice on EAF steelmaking suggest that the viscosity of molten steel is reduced, the velocity of molten steel is faster, and the stirring effect of bottom-blowing is better under the same bottom-blowing conditions, which is contrary to the results of numerical simulation. To further analyze and explain this phenomenon, physical model verification was conducted.

4.2.1. Physical Model Verification Plan

The water model experiments are widely used in the validation of numerical simulation in molten bath fluid flow [31,33,34] and achieve better effect. Generally, the KCl solution was added simultaneously with bottom-blowing gas during the water model experiment process. The effect of bottom-blowing stirring was expressed by the mixing time, which was recorded when the conductivity difference between the two conductivity electrodes installed in the model bottom below 5%. However, the physical model verification, showed in Figure 6, was quite different from the above.

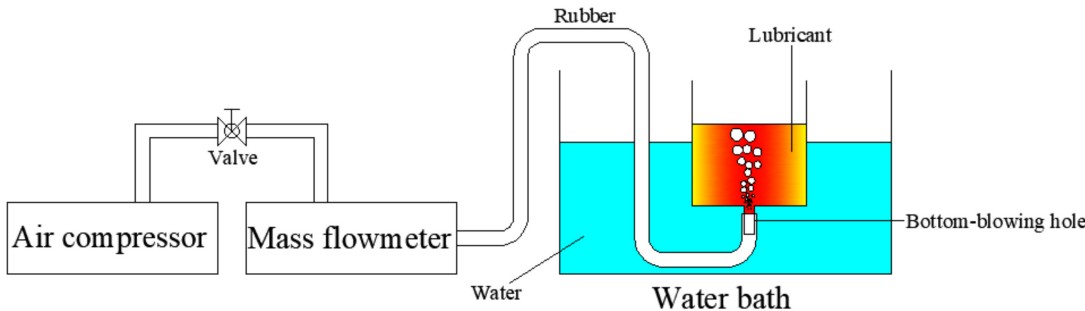

**Figure 6.** Connection diagram of the experimental device for physical model verification.

A transparent acrylic plexiglass tube mold, shown in Figure 7, with a cylinder at a diameter of 12 cm and a height of 10 cm in the upper part, and a circular bottom-blowing hole at a diameter of 1 cm and a height of 3 cm in the nether part, was used to simulate the effect of the EAF. The No. 11 lubricating oil inside the mold, produced by Mobil Glygoyle, was used to simulate the molten steel. It has stable performance and the relationship of the viscosity and density with temperature in the range of 0−100 °C was shown in Figure 8. The temperature characteristics of the different stages of smelting were simulated by adjusting the temperature of the water bath. The main instruments and parameters of the experiment are shown in Table 4.

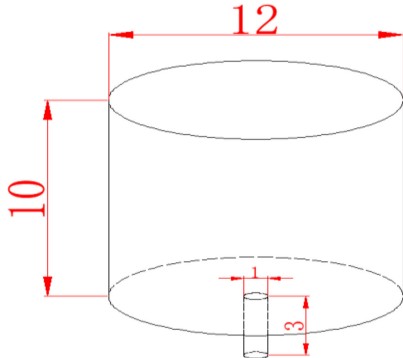

**Figure 7.** Schematic of the mold (cm).

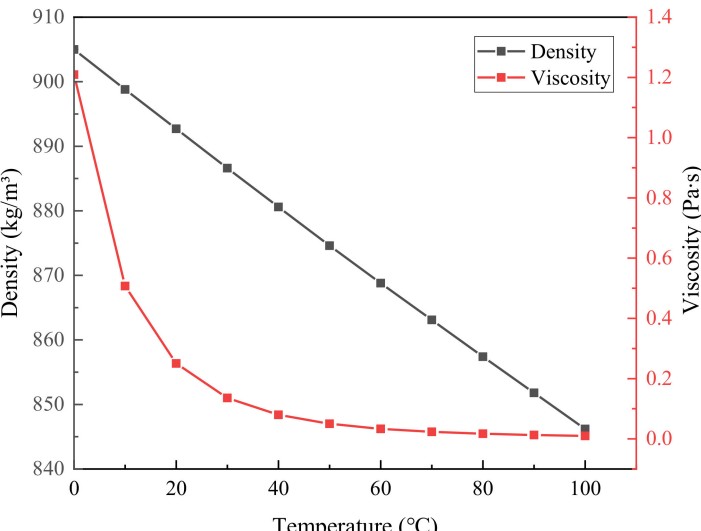

**Figure 8.** Changes in density and viscosity with the temperature of the No. 11 lubricating oil.

**Table 4.** Main instruments and parameters of physical model verification experiment.

| Instruments | Parameters |
| --- | --- |
| HH-2J magnetic stirring water bath | Power supply and heating power: 220 V 50 HZ, 600 W; Constant temperature range: room temperature 40–90 °C; Temperature accuracy: $\leq \pm 0.5$ °C; |
| ACO (Air Compression Operation) series electromagnetic air pump | Power supply and heating power: 220 VAC/ 50 HZ, 35 W; Displacement: 40 L/min |
| Mass flowmeter | Variable flow adjustment, the flow rate of this experiment is constant at 1 L/min. |
| Transparent acrylic plexiglass tube mold | Wall thickness: 5 mm |

To ensure the flows in physical model were similar to those in a real EAF, dynamic similarity between the two systems was determined based on the Froude number similarity criterion, shown as the following Equation (11) [15]:

$$\frac{Q_W}{Q_E} = \sqrt{\frac{d_W^2 D_W^2 H_W \rho_{lW} \rho_{gW}}{d_E^2 D_E^2 H_E \rho_{lE} \rho_{gE}}} \tag{11}$$

$$D_E = \frac{\pi \left( \frac{D_1}{2} \times \frac{A}{360} + \frac{D_2}{2} \times \frac{B}{360} \right) + 2L_1}{\pi} \tag{12}$$

where parameters with subscript W are those of physical model; parameters with E represent those of the EAF prototype; $Q_W$ and $Q_E$ are the gas flow rates of the physical model and the EAF prototype, m³/h; $d_W$ and $d_E$ are the nozzle diameters of the physical model and the EAF prototype, m; $H_W$ and $H_E$ are the molten bath depth of the physical model and the EAF prototype, m, as 80 and 1650 mm, respectively; $\rho_{lW}$ and $\rho_{lE}$ are the liquid density of the physical model and the EAF prototype, kg/m³; $\rho_{gW}$ and $\rho_{gE}$ are the gas density of the physical model and the EAF prototype, kg/m³; $D_W$ and $D_E$ are the molten bath hydraulic diameter of the physical model and the EAF prototype, m. The value of $D_W$ is 10 mm, while $D_E$ can be calculated by Equation (12) [15]. Where $A$ and $B$ are the actual fan angles at the top of the EAF prototype, as 110° and 233°, respectively; $D_1$ and $D_2$ are the diameter of the reference circle on the top surface of the eccentric area and the diameter

of the reference circle on the top surface of the main body of the EAF, as 1374 and 4072 mm, respectively; and $L_1$ is the length of the hypotenuse, as 2690 mm.

Before the experiment process, the flow rate of the mass flowmeter was set to 1 L/min, and it was confirmed that the connecting pipe was not leaking oil or gas. The water bath was warmed to 40 °C, and the temperature was maintained for 15 min. During the experiment process, an electromagnetic air pump was used to continuously blow the gas for 1 min. Bubbles were formed in the lubricating oil in the mold. The time required for the last bubble to rise to the liquid level was recorded. When the heat-transfer equilibrium reached the balanced state between the lubricating oil in the mold and the water in the magnetic stirring water bath, it can be considered that the temperature of the oil was equal to the water temperature (the external heat loss was ignored). The temperatures of the water bath pot set in the experiment were 40, 50, 60, 70, 80, and 90 °C, respectively. A group of bottom-blowing gas experiments was performed for each 10 °C, and each group was repeated seven times.

### 4.2.2. Physical Model Verification Results and Analysis

We recorded the time required from the start of the bottom-blowing operation to the rising of the last bubble in the lubricating oil to the liquid level in each test. The final data are shown in Table 5.

**Table 5.** Time of bubbles rising in the No. 11 lubricating oil at different temperatures.

| Temperature | Number of Experiments | | | | | | | Average Value (s) |
|---|---|---|---|---|---|---|---|---|
| | 1 | 2 | 3 | 4 | 5 | 6 | 7 | |
| 40 °C | 66.40 | 66.55 | 66.63 | 64.46 | 65.26 | 64.72 | 65.94 | 65.77 |
| 50 °C | 64.99 | 65.31 | 65.40 | 65.10 | 64.65 | 65.48 | 65.42 | 65.24 |
| 60 °C | 65.49 | 64.31 | 64.53 | 64.24 | 64.46 | 65.36 | 64.23 | 64.58 |
| 70 °C | 64.43 | 64.03 | 64.06 | 64.33 | 63.62 | 63.53 | 63.56 | 63.92 |
| 80 °C | 63.05 | 63.07 | 63.25 | 62.86 | 63.44 | 63.17 | 60.28 | 63.08 |
| 90 °C | 62.79 | 63.07 | 62.46 | 62.27 | 63.21 | 62.87 | 62.86 | 62.81 |

To ensure the accuracy and reliability of the data and improve the representative role of the average time in the sample data, the maximum and minimum values of the bubble escape time in each group were removed, and the rest of data were statistically verified.

The average value, variance between samples, and standard deviation of sample group were calculated by Equations (13)−(15), respectively.

$$\bar{t} = \frac{1}{n} \sum_{i=1}^{n} t_i \tag{13}$$

$$D(t) = \frac{1}{n-1} \sum_{i=1}^{n} (t_i - \bar{t})^2 \tag{14}$$

$$S(t) = \sqrt{\frac{1}{n-1} \sum_{i=1}^{n} (t_i - \bar{t})^2} \tag{15}$$

where $t$ is the bubble floating time, s; $\bar{t}$ is the average floating time of experimental bubbles in each group, s; $n$ is the number of samples in each group, $n = 5$; $D(t)$ is the variance of each group; and $S(t)$ is the standard deviation of each group.

Under the condition that the confidence is 95%, the confidence interval of each group of samples is calculated by Equation (16).

$$\left( \bar{t} - \frac{S(t)}{\sqrt{n}} Z_{\frac{\alpha}{2}}, \bar{t} + \frac{S(t)}{\sqrt{n}} Z_{\frac{\alpha}{2}} \right) \alpha \tag{16}$$

In the formula: $\alpha$ is the area surrounded by the continuous random variable, and its distribution density function and the $\alpha-$ quantile of this distribution, $\alpha = 0.05$; $Z_{\frac{\alpha}{2}}$ is the $\alpha-$ subject to the normal distribution quantile, $Z_{\frac{\alpha}{2}} = 1.96$.

The statistically verified experimental results are listed in Table 6. From this table, it is known that there is a 95% confidence that the bubble escape time of the bottom-blowing gas was located in the confidence interval at the corresponding set temperature, and as the temperature increased, the confidence intervals of the upper and lower bubble escape time limits decreased. Thus, the bubble escape time data in the physical model verification are reliable.

**Table 6.** Statistical verification results.

| Temperature | Variance | Standard Deviation | Confidence | Confidence Interval |
|---|---|---|---|---|
| 40 °C | 0.599 | 0.774 | 0.95 | (65.096, 66.452) |
| 50 °C | 0.036 | 0.190 | 0.95 | (65.077, 65.411) |
| 60 °C | 0.203 | 0.451 | 0.95 | (64.185, 64.975) |
| 70 °C | 0.105 | 0.324 | 0.95 | (63.636, 64.204) |
| 80 °C | 0.022 | 0.147 | 0.95 | (62.951, 63.209) |
| 90 °C | 0.049 | 0.222 | 0.95 | (62.616, 63.004) |

According to the data in Table 5, the relationship between the bottom-blowing bubble floating time and the lubricating oil temperature is plotted in Figure 9. It can be seen from Figures 8 and 9 that under the total gas volume of 1 min bottom-blowing, as the temperature increased, the viscosity of the No. 11 lubricating oil decreased, and the bubble escape time was shortened accordingly. The experimental results confirm the conclusion of numerical simulation, that is, with temperature increase, under the same bottom-blowing stirring conditions, the viscosity of the molten fluid decreases, making it more difficult to be stirred [35], and the bottom-blowing efficiency continues to decrease with the smelting process.

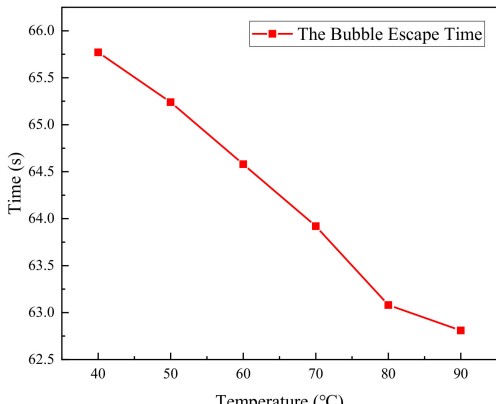

**Figure 9.** Relationship between bottom-blown bubble escape time and temperature.

To further explain this phenomenon, the relationship between argon and molten steel can be simplified as shown in Figure 10.

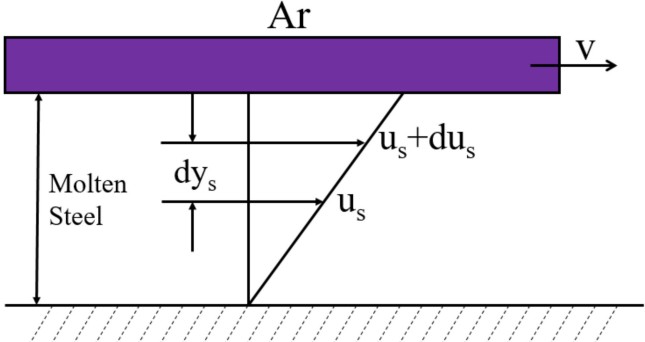

**Figure 10.** Schematic diagram of the interaction between argon and molten steel.

The bottom-blowing gas has a certain initial velocity and relies on the viscous force between the bottom-blowing gas and the molten steel to drive the flow of the molten steel. The argon gas close to the molten steel is in a laminar flow state, subject to Newton's viscosity theorem, as shown in Equation (17). According to Newton's viscosity theorem, the magnitude of the internal friction stress $\tau$ is proportional to the velocity difference $du_s$ between the two fluid layers and inversely and the distance $dy_s$ between the two fluid layers. Due to the same conditions used, $du_s/dy_s$ is a constant, set to $k_s$.

$$\tau = \mu \frac{du_s}{dy_s} = \mu k_s \tag{17}$$

In the formula: $\tau$ is the internal friction shear stress between the two steel liquid layers, N; $\mu$ is the viscosity of the steel liquid, Pa·s;

The viscous force is the resistance for argon, and for the molten steel, the internally generated viscous force is the power of the molten steel and obeys Newton's second law. The relationship between speed, acceleration, and time is shown in Equation (20). Equation (20) can be obtained by combining Equations (17)−(19). The flow rate of molten steel is proportional to the viscosity and time.

$$\tau = ma \tag{18}$$

$$v = at \tag{19}$$

$$v = \frac{\mu k_s}{m} t \tag{20}$$

where $a$ is the acceleration of the molten steel flow layer, m/s$^2$; $m$ is the mass of the molten steel flow layer, kg; $v$ is the flow rate of the molten steel, m/s; and $t$ is the acceleration time of the molten steel, s.

From Equation (20), we can know that under the same bottom-blowing conditions, the farther away from the steel–slag contact surface, the longer the distance the molten steel is accelerated, and the greater the flow rate of the molten steel. In addition, with the progress of EAF smelting, the viscosity of the molten steel decreased, the acceleration effect of the bottom-blowing stirring on the molten steel worsened, and the flow rate of the molten steel decreased.

### 4.3. Operation System Discussion for Bottom-Blowing

Based on the simulation results and related analysis of the characteristic points of the four representative stages of EAF steelmaking, the four characteristic points selected can reflect the characteristics of the four representative stages. Combining simulation results with the physical model verification results, we suggest that the bottom-blowing flow control can be optimized according to bottom-blowing efficiency and smelting actual needs studied in this paper. The specific process can be controlled in four stages.

(1) Stage one represents the smelting period at the end of the smelting feed. In this stage, the melt pool has the highest viscosity, the bottom-blowing efficiency is high, and the power supply intensity is the largest. It is necessary to accelerate the melting of scrap steel. Therefore, using a higher bottom-blowing flow rate can accelerate the heat transfer of the melt pool without consuming a large amount of bottom-blowing gas and melting scrap;

(2) Stage two represents the early stage of decarburization in the smelting process. In this stage, the intensity of oxygen supply is large, and the internal chemical reaction in the molten pool is intensely exothermic. In addition to the stirring effect of the oxygen jet on the molten pool, CO produced by the decarburization reaction also has a larger mixing effect [36,37]. Combined with the fact that the molten steel has a higher viscosity and bottom-blowing efficiency, it is recommended to use a lower bottom-blowing flow rate to bring the molten steel to a better flow state at a lower flow rate;

(3) Stage three represents the mid-to-late stage of decarburization in the smelting process. In this stage, the oxygen supply intensity is relatively less, the chemical reaction

in the molten pool gradually weakens, and CO produced by the decarburization reaction weakens the stirring effect of the molten pool and reduces the amount of heat. The viscosity of the molten steel and the bottom-blowing efficiency is low. To ensure a good stirring effect, it is recommended to use a moderate bottom-blowing flow;

(4) Stage four represents the final stage of smelting. In this stage, the decarburization reaction in the furnace is close to the end point, and local oxidation is easy to cause iron loss. Although the viscosity and the bottom-blowing stirring efficiency are the lowest, to prevent local over-oxidation, reduce the loss of iron, and improve the metal yield, it is recommended to use a higher bottom-blowing flow rate.

## 5. Conclusions

In this paper, the industrial 100 t EAF was used as the reference model; the numerical simulation software was used to establish a three-dimensional grid model with 55,924 grids and 540,395 nodes; the fluid calculation software ANSYS Fluent 18.2 was used to calculate the velocity field of the molten steel under the same bottom-blowing gas flow rate at different EAF smelting periods; and the fluidity and bottom-blowing effect of the molten steel at different smelting stages were studied. The conclusions are summarized as follows:

(1) When the bottom-blowing gas flow rate was 100 L/min, the average flow rates of the four stages in EAF steelmaking were $v_1$ = 0.0081 m/s, $v_2$ = 0.0069 m/s, $v_3$ = 0.0063 m/s, and $v_4$ = 0.0053 m/s. Numerical simulation results showed that under the same bottom-blowing conditions, the longer the distance from the steel–slag contact surface, the greater the flow rate of the molten steel.

(2) The following relationship between the molten steel viscosity and flow rate was obtained: $v = -85048.18\eta^2 + 1808.75\eta - 1.473$. With the progress of the smelting stage, the viscosity of molten steel decreased, and flow velocity of the molten steel decrease.

(3) Physical model verification results indicated when the bottom-blowing stirring conditions remain unchanged, the temperature of the No. 11 lubricating oil increased, the viscosity of it decreased, and the bubble escape time would decrease. This confirmed the conclusion of numerical simulation, that is, with temperature increase, under the same bottom-blowing stirring conditions, the viscosity of the molten fluid decrease. The kinetic energy exchange between the bottom gas and the molten steel became worse, and the acceleration effect driven by bottom-blowing gas worsened, making the molten steel more difficult to be stirred, meaning the bottom-blowing efficiency continued to decrease with the smelting process.

(4) According to the numerical simulation and physical model verification results, relevant suggestions are made for the operation system of bottom-blowing in EAF steel-making. The higher bottom-blowing flow rates should be used during the smelting period and the end of smelting period while the lower bottom-blowing flow rates can be used in the early stage of decarburization and moderate bottom-blowing flow rates used in the middle and later stages of decarburization.

**Author Contributions:** Conceived of and designed the experiments, L.Y., Y.G., F.C., S.W., and H.H.; performed the experiments, H.H. and L.Y.; analyzed the data, H.H., F.Z.; searched the relevant literature and data, H.H. and B.L.; wrote the paper, H.H.; reviewed and contributed to the final manuscript, L.Y. and Y.G. All authors have read and agreed to the published version of the manuscript.

**Funding:** This research was funded by the National Natural Science Foundation of China (No. 51804345) and the Fundamental Research Funds for the Central Universities of Central South University (No. 2020zzts752).

**Institutional Review Board Statement:** Not applicable.

**Informed Consent Statement:** Not applicable.

**Data Availability Statement:** Restrictions apply to the availability of these datas. Datas were obtained from Central South University and are available from the Hang Hu with the permission of Central South University.

**Acknowledgments:** Financial support from the National Natural Science Foundation of China (No. 51804345) and the Fundamental Research Funds for the Central Universities of Central South University (No. 2020zzts752) is gratefully acknowledged.

**Conflicts of Interest:** The authors declare no conflict of interest.

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
