# Peer review of "Numerical Simulation of Bottom-Blowing Stirring in Different Smelting Stages of Electric Arc Furnace Steelmaking"

_metals, doi:10.3390/met11050799_

Round 1

Reviewer 1 Report

Dear Authors,

Many thanks for your submitted work. I can see you did a great job but at some points you couldn't represent what you really did. Please find below comments on your work:

  • First of all the English grammar is very poor and typos are visible everywhere. As a sample, I can say you used long sentences and also some words are repeated twice in a sentence! like in line 16 "the properties of the properties and ... and ...."
  • I couldn't see any novel findings in your work! Could you clearly mention what is your novelty! 
  • How you connect your experimental case with the real case! As you may know there are plenty works with water models and tracers as well. Your work is very similar to them! 
  • Your references are very old! As far as I know there are more papers published recently in Journals and you can find them! 
  • How you connect your prototype with the real case! I could expect you compare some dimensionless number! 
  • Please revise your introduction and make it updated with recently published paper and as a sample please go to some more details of them! 
  • You have presented only one contours from CFD? I need to see more contours and more clear ones! Please at least magnify them! 
  • I could not see that you described your model and experimental rigs too many details have been missed! 
  • I searched your paper and couldn't find the mesh dependency check! where is it? Are you sure your results are very less dependent on the mesh sizes!
  • Please always be aware of using ANSYS! once you name it you need to properly mention the details regarding how you access them and proper reference must be added to the reference lists!  

At current form I suggest you resubmit it after changes and at present format is not suitable for publication in Metals.

Author Response

Dear Reviewer:
We are deeply thankful for the professional opinions and kind suggestions that have greatly improved our manuscript. With your careful work and warm reminders, under the discussion and negotiation with all the coauthors, we have made the following corrections to the manuscript:

Reviewer 2 Report

Dear Authors,

your paper "Numerical Simulation of Bottom Blowing Stirring in Different Smelting Periods of Electric Arc Furnace Steelmaking" aims to highlight the loose of stirring efficiency along the EAF smelting process by maintaining constant the stirring conditions promoved by botton gas stirring procedure.

Even the topic is really interesting and can have significant industrial implications, the quality of your paper is poor. Not for the contents, that are quite innovative, but for the style of presentation.

The different sections are confusing, excessively long and written in a poor english. There are a lot of grammar errors (especially in punctuation) and typo.

In particular, Intoduction needs a revision about the description of the operational steps in an EAF. Generally, after the scrap charging, melting is the first operation. From two to four scrap buckets are charged as a function of steel grade and scrap quality. Only after the last bucket charged, the refining period starts. When the temperature is still relatively low, i.e., at the beginning of the refining stage (completely molten pool), dephosporization takes place. When the temperature increases, decarburization takes place, and it continuos until the C endopoint is reached. During this period, oxygen is blown to oxidize the bath but also carbon and oxygen are injected at the slag level to promote the foaming slag. When the setpoint is reached (°C, C% and P%), oxygen is stopped and steel is poured into a ladle. All the reducing operation are today performed outside the furnace, in the so-called "secondary metallurgy". Thus, I suggest to strongly revise the introduction part by the support of a steelmaker, to better describe the steelmaking process in an EAF

In addition, excess of simplifications were made to decribe the physics of steel and slag. Steel chemical composition was not considered, as well the slag was considered as a liquid with constant viscosity. Slag is a complex system and viscosity vary in a wide range as a function of temperature and chemical composition. In my opinion, these aspects are fundamental to reach a high accuracy simulation.

The discussion is excessively complicated and not clarify the behavior observed by the Authors thorugh the simulation. A poor comparison with the literature is also made.

Thus, I cannot recommend you paper for publication.

Nevertheless, I attach some comments directly made in the PDF files, to improve the quality of your paper and try a resubmission.

Finaly I strongly suggest an english style revision

Best Regards

Author Response

Dear Reviewer:

We are deeply thankful for the professional opinions and kind suggestions that have greatly improved our manuscript. With your careful work and warm reminders, under the discussion and negotiation with all the coauthors, we have made the following corrections to the manuscript:

Author statements to initial comments:

Reviewer #2 (Remarks to the Author)

Dear Authors,

your paper "Numerical Simulation of Bottom Blowing Stirring in Different Smelting Periods of Electric Arc Furnace Steelmaking" aims to highlight the loose of stirring efficiency along the EAF smelting process by maintaining constant the stirring conditions promoved by bottom gas stirring procedure.

Even the topic is really interesting and can have significant industrial implications, the quality of your paper is poor. Not for the contents, that are quite innovative, but for the style of presentation.

We are deeply grateful to the reviewer for the critiques and suggestions that have greatly improved our manuscript.

The different sections are confusing, excessively long and written in a poor english. There are a lot of grammar errors (especially in punctuation) and typo.

We really apologize for making these mistakes in English grammar and usage. We have read and carefully checked the text for any grammatical or spelling mistakes. The correction are highlighted in red in manuscript.

In particular, introduction needs a revision about the description of the operational steps in an EAF. Generally, after the scrap charging, melting is the first operation. From two to four scrap buckets are charged as a function of steel grade and scrap quality. Only after the last bucket charged, the refining period starts. When the temperature is still relatively low, i.e., at the beginning of the refining stage (completely molten pool), dephosporization takes place. When the temperature increases, decarburization takes place, and it continuous until the C endpoint is reached. During this period, oxygen is blown to oxidize the bath but also carbon and oxygen are injected at the slag level to promote the foaming slag. When the setpoint is reached (°C, C% and P%), oxygen is stopped and steel is poured into a ladle. All the reducing operation are today performed outside the furnace, in the so-called "secondary metallurgy". Thus, I suggest to strongly revise the introduction part by the support of a steelmaker, to better describe the steelmaking process in an EAF

In addition, excess of simplifications were made to describe the physics of steel and slag. Steel chemical composition was not considered, as well the slag was considered as a liquid with constant viscosity. Slag is a complex system and viscosity vary in a wide range as a function of temperature and chemical composition. In my opinion, these aspects are fundamental to reach a high accuracy simulation.

We are grateful to the reviewer by this great suggestion. We introduced the superiority of electric arc furnace steelmaking and the application of bottom-blowing stirring system, then pointed out some disadvantages under bottom-blowing gas flow rate at different EAF smelting periods, such as insufficient stirring intensity, the scouring of the furnace lining. We found that determining the main flow properties of molten steel in different stages and studying the effect of bottom-blowing stirring on molten steel could help understand the bottom-blowing effect in different stages of the electric arc furnace smelting process, and strengthened the mixing effect of molten steel.

The operational steps in an EAF could be obtained in section 2.2. After introduced the different operational steps in EAF steelmaking, we combined the thermal system of steelmaking process, the temperature and chemical composition of molten steel and divided the smelting process of the EAF into four stages. But it is of no problem that the contents about EAF operational steps should be moved to Introduction if necessary.

The discussion is excessively complicated and not clarify the behavior observed by the Authors through the simulation. A poor comparison with the literature is also made.

We apologize for not making this information clearer. Our idea is to simulate the velocity field of the molten steel in different EAF melting stages under the same bottom-blowing gas flow rates by ANSYS Fluent 18.2. The results of the numerical simulation were quite different from what we had expected. To verify the numerical simulation results, we performed physical model verification and obtained the same results, that is the temperature of molten steel rises and the viscosity of molten steel decreases with the ongoing smelting process, under the condition that the bottom-blowing gas flow rate remained unchanged, the average molten steel flow rate decreased significantly. Which means the effect of bottom blowing stirring becomes worse. In section 4.3, we can explain this phenomenon well by theoretical analysis. The correction are highlighted in red in manuscript.

Thus, I cannot recommend your paper for publication.

Nevertheless, I attach some comments directly made in the PDF files, to improve the quality of your paper and try a resubmission.

Finally I strongly suggest an English style revision

Best Regards

We are grateful for the reviewer’s careful work and valuable comments. We have made great revision in manuscript.

Looking forward to your decision,

With kind regards,

Sincerely yours,

Mr Hang Hu

School of Minerals Processing and Bioengineering, Central South University, Changsha 410083.

E-mail address: [email protected]

Reviewer 3 Report

Quite an interesting study, but with some major issues that need resolving.

  1.  In places it is quite difficult to follow both the experimental methodology and the hypotheses/explanations of the results because of the rather poor quality of the English language. This will need to be improved
  2. The explanation of the relationship between the gas flow rate, flow velocity, viscosity and blowing efficiency is very confusing in places. Section 4.2 is a very good example of this confusing explanation. This relationship needs to be better explained in order to understand the effect of each individual factor on the observed results. How is bottom blowing efficiency calculated ?
  3. It is not entirely clear to me how the results from the physical modelling validate the results from the simulations - I can appreciate the basic theory, but this relationship needs to be much better explained.
  4. The novelty of the results is not clear - the authors need to identify what is novel about this research.
  5. There are some key experimental details that are either missing or not clear - what is Number 11 Lubricating Oil ? What is the composition and manufacturer ?
  6. There are many examples where the methodology used reads like an instruction manual - lines 362- 375 for example. These need to be removed and rewritten
  7. The data in Table 1 are key to the simulations. It is not clear where these results have been obtained or how they have been calculated. Ref.26 is mentioned but this just refers to a Steelmaking Manual. How accurate is this data? Are there any other other sources of compositional and viscosity data that could be used to represent the four different stages that the authors have used in this study?
  8. Section 4.4 is the first time that the effect of CO from decarburisation is mentioned. The effect of CO is discussed in 2 of the recommendations (2&3) in this section. "CO produced has a larger mixing effect ..." and "CO weakens the stirring effect of the molten pool and reduces the amount of heat... " This needs further clarification or references need to be supplied for the explanations that have been offered in this section. In fact there are lots of claims in section 4.4 that need referencing. Iron oxidation, iron loss etc
  9. Conclusion 3 needs rewriting - at present it is very confusing.

Author Response

Dear Reviewer:

We are deeply thankful for the professional opinions and kind suggestions that have greatly improved our manuscript. With your careful work and warm reminders, under the discussion and negotiation with all the coauthors, we have made the following corrections to the manuscript:

Author statements to initial comments:

Reviewer #3 (Remarks to the Author)

Quite an interesting study, but with some major issues that need resolving.

We greatly thank the reviewer for all the comments and additional suggestions that have greatly improved the manuscript.

  1. In places it is quite difficult to follow both the experimental methodology and the hypotheses/explanations of the results because of the rather poor quality of the English language. This will need to be improved

We really apologize for making these mistakes in English grammar and usage. We have read and carefully checked the text for any grammatical or spelling mistakes. The correction are highlighted in red in manuscript.

  1. The explanation of the relationship between the gas flow rate, flow velocity, viscosity and blowing efficiency is very confusing in places. Section 4.2 is a very good example5 of this confusing explanation. This relationship needs to be better explained in order to understand the effect of each individual factor on the observed results.

We apologize for this; we have edited the contents to make this clearer. The changes are highlighted in red in manuscript (lines 302-425).

How is bottom blowing efficiency calculated?

We thank the reviewer for raising this important point and apologize for not making this clearer in the manuscript. In this manuscript, we think that under the acceleration of bottom-blowing gas stirring, the greater the flow speed of molten steel, the better the bottom blowing mixing effect, and the higher the utilization rate of bottom blowing gas.

  1. It is not entirely clear to me how the results from the physical modelling validate the results from the simulations - I can appreciate the basic theory, but this relationship needs to be much better explained.

We apologize for not making this information clearer. The water model experiments are widely used in the validation of numerical simulation in molten bath fluid flow [27-29], the effect of bottom-blowing stirring was expressed by recording the mixing time when the conductivity difference between the two conductivity electrodes installed in the model bottom below 5%. In the text, the flow dynamic similarity between the physical model and the EAF prototype was determined based on the Froude number similarity criterion. In a real EAF, under the same bottom blowing conditions, with the ongoing smelting process, the temperature of molten bath increased, the viscosity of molten steel decreased, and the velocity of molten steel decreased, which meant the acceleration effect of the bottom-blowing stirring on the molten steel worsened and insufficient exchange of kinetic energy between bottom-blowing gas and molten steel. In the physical model verification, under the same bottom blowing conditions, with temperature of the No. 11 lubricating oil increased, the viscosity of the molten fluid decreased, making it more difficult to be stirred, so the bubble escape time was shortened due to insufficient exchange of kinetic energy between bottom-blowing gas and No. 11 lubricating oil, the bottom-blowing efficiency decreased. The two situations are essentially the same in fluid flow under the bottom-blowing stirring.

4.The novelty of the results is not clear - the authors need to identify what is novel about this research.

We apologize for not making this information clearer. Our numerical simulation results showed that with the ongoing smelting process, the temperature of molten steel rises and the viscosity of molten steel decreases. Under the condition that the bottom-blowing gas flow rate remained unchanged, the average molten steel flow rate decreased significantly. Which means the effect of bottom blowing stirring becomes worse. However, research on EAF steelmaking suggests that the viscosity of molten steel is reduced, the velocity of molten steel is faster, and the stirring effect of bottom -blowing is better under the same bottom-blowing conditions. Viscosity is a physical quantity that characterizes the difficulty of fluid flow. When the viscosity of fluid is small, the resistance of fluid flow is small, and it is easier to move under the action of external force. So physical model verification experiment was conducted and we achieved the same results. Through further theoretical analysis, we obtained the relationship between the flow velocity of molten steel with the viscosity, quality of molten steel, and the time of molten steel accelerated, which was showed as . With this equation, we can know that bottom-blowing gas flow rate and quality of molten steel remained unchanged, when the viscosity of molten steel and the time of molten steel accelerated is small, the flow velocity of molten steel is small.

In brief, we divided the smelting process of the EAF into four stages and found anomalous phenomenon by numerical simulation, that was when the viscosity of molten steel decreased, under the same bottom blowing conditions, the velocity of molten steel decreased. Which is contrary to the artificial experience and industrial production practice. Then we designed the physical model verification and explained the anomalous phenomenon. The physical model verification results showed that under the same bottom blowing conditions, the viscosity of molten fluid decreased, the fluid velocity decreased, the acceleration effect driven by bottom-blowing gas worsen, making the molten fluid more difficult to be stirred, so the bottom-blowing efficiency continued to decrease with the smelting process.

5.There are some key experimental details that are either missing or not clear - what is Number 11 Lubricating Oil ? What is the composition and manufacturer ?

We apologize for this oversight. The No. 11 lubricating oil produced by Mobil Glygoyle, was widely used in gears and bearings due to its strong heat and oxidation resistances. Now we have added the contents about the No 11 Lubricating Oil. For the reviewer’s convenience the specific changes to the manuscript text are shown below (lines 325-328, changes highlighted in red):

A transparent acrylic plexiglass tube mold, shown in Figure 7, with a cylinder at a diameter of 12 cm and a height of 10 cm in the upper part, and a circular bottom-blowing hole at a diameter of 1 cm and a height of 3 cm in the nether part, was used to simulate the effect of the EAF. The No. 11 lubricating oil inside the mold, produced by Mobil Glygoyle, was used to simulate the molten steel. It has stable performance and the relationship of the viscosity and density with temperature in the range of 0~100 ℃ was shown in Figure 8. The temperature characteristics of the different stages of smelting were simulated by adjusting the temperature of the water bath. The main instruments and parameters of the experiment are shown in Table 4.

6.There are many examples where the methodology used reads like an instruction manual - lines 362- 375 for example. These need to be removed and rewritten

We are grateful to the reviewer for pointing this out as this is likely the most correct interpretation. We have modified and rewritten the contents in the manuscript. For the reviewer’s convenience the specific changes to the manuscript text are shown below (lines 351-362, changes highlighted in red):

Before the experiment process, the flow rate of the mass flowmeter was set to 1 L/min, and it was confirmed that the connecting pipe was not leaking oil or gas. The water bath was warmed to 40 °C, and the temperature was maintained for 15 min. During the experiment process, an electromagnetic air pump was used to continuously blow the gas for 1 min. Bubbles were formed in the lubricating oil in the mold. The time required for the last bubble to rise to the liquid level was recorded. When the heat transfer equilibrium reached the balanced state between the lubricating oil in the mold and the water in the magnetic stirring water bath, it can be considered that the temperature of the oil was equal to the water temperature (the external heat loss was ignored). The temperatures of the water bath pot set in the experiment were 40 °C, 50 °C, 60 °C, 70 °C, 80 °C, and 90 °C, respectively. A group of bottom blowing gas experiments was performed for each 10°C, and each group was repeated seven times.

7.The data in Table 1 are key to the simulations. It is not clear where these results have been obtained or how they have been calculated. Ref.26 is mentioned but this just refers to a Steelmaking Manual. How accurate is this data? Are there any other other sources of compositional and viscosity data that could be used to represent the four different stages that the authors have used in this study?

We thank the reviewer for raising this important point. Based on the smelting characteristics of EAF steelmaking process, we combined the thermal system of steelmaking process, the temperature and chemical composition of molten steel and divided the smelting process of the EAF into four stages. Then we referred to the calculation method of reference [19] and [26] to obtain the viscosity of molten steel in four stages.

8.Section 4.4 is the first time that the effect of CO from decarburisation is mentioned. The effect of CO is discussed in 2 of the recommendations (2&3) in this section. "CO produced has a larger mixing effect ..." and "CO weakens the stirring effect of the molten pool and reduces the amount of heat... " This needs further clarification or references need to be supplied for the explanations that have been offered in this section. In fact there are lots of claims in section 4.4 that need referencing. Iron oxidation, iron loss etc

We thank the reviewer for pointing this out. As requested, we have now added references in the corresponding position. For the reviewer’s convenience the specific changes to the manuscript text are shown below (line 450, changes highlighted in red):

  1. Conclusion 3 needs rewriting - at present it is very confusing.

We are grateful to the reviewer by this great suggestion. We have modified and rewritten the contents. For the reviewer’s convenience the specific changes to the manuscript text are shown below (lines 465-495, changes highlighted in red):

  1. Why the Authors have described the effect of alloying elements on viscosity if only the equation (1) was used to determine the steel viscosity at the four stages? Why the authors did not correct the Roscoe viscosity by the chemical composition to be more accurate?

We thank the reviewer for pointing this out. We have now deleted the contents about the effect of alloying elements on viscosity.

After consultation and communication with coauthors, we thought that equation (1) in reference [19] was relatively accurate compared with other calculating method, and the difference between the results and reference 27 is very small.

  1. Why the carbon content of the charge is so high? High C-DRI or pig-iron is charged togheter steel scraps? Please, clarify!

We apologize for not making this information clearer. Actually, hot metal is often added in EAF in China. The carbon content in hot metal is higher than that in scrap. And we choose the EAF with high hot metal ratio, so the temperature of molten iron is of quite difference during EAF steelmaking process, and the effect of bottom- blowing can be better showed, the numerical simulation results will be more obvious.

  1. From your results, it seems that the steel is stirred slower near the heart while it is stirred faster at the slag-steel interface. However, the bottom porous plugs are sited in the heart. Could the Authors explain why the stirring effect is higher at the slag-steel interface and not in the proximity of the injection point?

We are grateful to the reviewer for pointing this out. From the equation (20), we can know that under the same bottom blowing conditions, the farther away from the steel-slag contact surface, the longer the distance the molten steel is accelerated, and the greater the flow rate of the molten steel. In addition, with the progress of EAF smelting, the viscosity of the molten steel decreased, the acceleration effect of the bottom-blowing stirring on the molten steel worsened, and the flow rate of the mol-ten steel decreased.

Looking forward to your decision,

With kind regards,

Sincerely yours,

Mr Hang Hu

School of Minerals Processing and Bioengineering, Central South University, Changsha 410083.

E-mail address: [email protected]

Round 2

Reviewer 1 Report

Dear Authors,

Many thanks for your efforts. Now the paper have been improved a lot. It has reasonable quality for Metals.

Author Response

Dear Reviewer:

We are deeply thankful for the professional opinions and kind suggestions that have greatly improved our manuscript. With your careful work and warm reminders, under the discussion and negotiation with all the coauthors, we have added some reference and made minor corrections to the manuscript (changes highlighted in red). The final version of statements is showed in the attachments.

Looking forward to your decision,

With kind regards,

Sincerely yours,

Mr. Hang Hu.

School of Minerals Processing and Bioengineering, Central South University, Changsha 410083.

E-mail address: [email protected]

Reviewer 2 Report

Dear Authors,

thank you for submitting a revised version of your paper.

After a careful check, it is possible to appreciate the efforts to improve the quality of presentation and the scientific soundness.

The paper can be suitable to publication in Metals after minor revisions, listed below.

Abstract

line 12-13: “intermediate and final smelting stage, and the ending smelting stage”. I cannot distinguish final smelting stage from ending smelting stage. Maybe it would be more correct say: intermediate smelting stage, and the ending smelting stage. Please check

Introduction

line 58, 59, 60: masspct. Please change to wt.%. Change pct to wt.%. Please modify also along the whole text

Flow characteristics of molten steel in different smelting periods of EAF

line 95: please explicit the acronym EBT

line 132: With the temperature increase. Please replace as “with the temperature increase”

line 140: Silicon and manganese. Please replace as “silicon and manganese”

Table 1: could the Authors add a brief justification about the initial C concentration of the bath (2.581%) through the description of the scrap mix

Line 204: “A 100t industrial EAF in a steel plant”. Is a top-charge EAF or a continuous-charge EAF?

Numerical simulation of bottom-blowing

Line 237-328: “viscosity of the steel slag 237 to 0.3 Pa·s.”. Why it was set ti 0.3 Pa.s?

The viscosity of the slag depends upon several factor, in particular: chemical composition, temperature and solid fraction. The viscosity can be well estimated by the Einstein-Roscoe equation, that consider the solid mass fraction and correct the real viscosity of the liquid slag. I suggest having a look on this paper:

FUNDAMENTALS OF EAF AND LADLE SLAGSAND LADLE REFINING PRINCIPLES By Eugene Pretorius and Baker Refractories to implement the viscosity model into your model

Results and discussion

Line 395-402: Please, support your conclusion with a suitable reference relying on the relationship between drainage bubble speed and liquid viscosity. For instance, critical evaluation of role of viscosity and gas flowrate on slag foaming by Barella et al. or similar works.

Best Regards

Author Response

Dear Reviewer:

We are deeply thankful for the professional opinions and kind suggestions that have greatly improved our manuscript. With your careful work and warm reminders, under the discussion and negotiation with all the coauthors, we have made the following minor corrections to the manuscript (changes highlighted in red):

Author statements to initial comments:

Reviewer #2 (Remarks to the Author)

Dear Authors,

Thank you for submitting a revised version of your paper. After a careful check, it is possible to appreciate the efforts to improve the quality of presentation and the scientific soundness. The paper can be suitable to publication in Metals after minor revisions, listed below.

We are extremely grateful to the reviewer for the positive review and for these comments. For the reviewer’s convenience we have addressed each comment/critique separately:

Abstract line 12-13: “intermediate and final smelting stage, and the ending smelting stage”. I cannot distinguish final smelting stage from ending smelting stage. Maybe it would be more correct say: intermediate smelting stage, and the ending smelting stage. Please check

We thank the reviewer for raising these points that have greatly enhanced the manuscript. For the reviewer’s convenience the specific changes to the manuscript text are shown below (lines 12-13, Table 1, changes highlighted in red):

To explore the effect of molten steel characteristics on bottom-blowing at various stages of smelting, we divided the smelting process of the EAF into four stages: the melting stage, the early decarburization stage, the intermediate smelting stage, and the ending smelting stage.

Introduction line 58, 59, 60: masspct. Please change to wt.%. Change pct to wt.%. Please modify also along the whole text

We apologize for not having unified the statements. In Wei’s study [14], with the bottom-blowing stirring, the conditions of dephosphorization and decarburization improved, the mass of phosphorus in the molten steel decreased. Compared to the case without bottom-blowing stirring, the dephosphorization and decarburization rate are, respectively, increased by 12.1 and 11.8 %, and the contents of FeO and T.Fe in endpoint slag were, respectively, reduced by 4.1 and 4.7 wt.%. Now we have used wt.% in place of masspct in the whole text. For the reviewer’s convenience the specific changes to the manuscript text are shown below (lines 58-60, changes highlighted in red):

Wei [14] studied the velocity distribution in the molten bath at different bottom-blowing gas flow rates, and revealed the velocity of molten steel increased when the bot-tom-blowing gas rates increased, the content of phosphorus in the molten steel was de-creased by 0.005 wt.%, the contents of FeO and T. Fe in endpoint slag were, respectively, reduced by 4.1 and 4.7 wt.%, the dephosphorization and decarburization rate are, respectively, increased by 12.1 and 11.8 %, and the endpoint carbon-oxygen equilibrium of the molten steel is improved by 0.0024.

Flow characteristics of molten steel in different smelting periods of EAF line 95: please explicit the acronym EBT

We apologize for not making this information clearer. Now we have added the explanation to the acronym "EBT" in line 95 (changes highlighted in red).

line 132: With the temperature increase. Please replace as “with the temperature increase”

We apologize for this oversight. Now we have modified it in the manuscript. (line 132, changes highlighted in red):

line 140: Silicon and manganese. Please replace as “silicon and manganese”

We apologize for our carelessness. Now we have modified it in the manuscript. (line 140, changes highlighted in red):

Table 1: could the Authors add a brief justification about the initial C concentration of the bath (2.581%) through the description of the scrap mix

We apologize for not making this information clearer. Actually, hot metal is often added in EAF in China. The carbon content in hot metal is higher than that in scrap. And we choose the EAF with high hot metal ratio, so the temperature of molten iron is of quite difference during EAF steelmaking process, and the effect of bottom- blowing can be better showed, the numerical simulation results will be more obvious.

In the EAF steelmaking process, before the charges (hot metal and scrap) were added to the EAF, the chemical composition of hot metal and scrap would be tested and analyzed and we could know C concentration in hot metal and scrap. Then we could obtained the initial C concentration of the bath by dividing the total mass of C by the total weight of molten steel. The initial content of the remaining elements, such as Silicon, were obtained in the same way as that of carbon.

For the reviewer’s convenience the specific changes to the manuscript text are shown below (lines 162-165, changes highlighted in red):

According to the various characteristics of elements and temperature in the EAF steelmaking process, four liquid steel components and temperatures were selected in this work as the characteristic points of the four smelting stages. The theoretical values of the liquid steel density and viscosity were calculated based on Equation (1) and Reference [30]. Before the EAF steelmaking process, the chemical composition of hot metal and scrap would be tested and analyzed. Then the initial carbon and silicon content of the bath could be obtained by dividing the total mass of carbon and silicon by the total weight of molten steel, respectively. The specific parameters are shown in Table 1.

Line 204: “A 100t industrial EAF in a steel plant”. Is a top-charge EAF or a continuous-charge EAF?

We apologize for not making this information clearer. In China, the top-charge EAF is still widely used in most EAF steelmaking plants. However, with the development of equipment and technology in EAF steelmaking, many newly-built EAF steelmaking plants have gradually started to use continuous-charge EAF. In this study, a 100t industrial top-charge EAF was selected to research the flow characteristics of molten steel with bottom-blowing stirring. Now we have modified the contents in the manuscript. For the reviewer’s convenience the specific changes to the manuscript text are shown below (lines 204, changes highlighted in red):

A 100t industrial top-charge EAF in a steel plant was selected to study the physical and chemical properties of molten steel in the EAF steelmaking process.

Numerical simulation of bottom-blowing

Line 237-328: “viscosity of the steel slag 237 to 0.3 Pa·s.”. Why it was set ti 0.3 Pa.s?

The viscosity of the slag depends upon several factor, in particular: chemical composition, temperature and solid fraction. The viscosity can be well estimated by the Einstein-Roscoe equation, that consider the solid mass fraction and correct the real viscosity of the liquid slag. I suggest having a look on this paper:

FUNDAMENTALS OF EAF AND LADLE SLAGSAND LADLE REFINING PRINCIPLES By Eugene Pretorius and Baker Refractories to implement the viscosity model into your model

We are grateful to the reviewer for pointing this out and providing literature reference. We fully agree with the reviewer's opinions that the composition of slag is very complicated and the viscosity of the slag, influenced by chemical composition, temperature and solid fraction, is usually not fixed, and when we study the properties of slag, we can not set the viscosity of slag to a constant value. However, in this paper, we focused on the velocity field of molten steel under the condition of bottom blowing-stirring in different stages in EAF steelmaking process. In the process of numerical simulation, after referring to the viscosity data of slag in reference [14] and [31], we decided to set the viscosity of slag to 0.35 Pa·s (We really apologize for our carelessness that we missed a number, we set the viscosity of slag to 0.35 Pa·s instead of 0.3 Pa·s), the simulation results were acceptable within the allowable error range. But just as the reviewer said, considering the complexity of the slag, and using the Einstein-Roscoe equation to estimate the viscosity of the slag, could get more precise simulation results.

Results and discussion

Line 395-402: Please, support your conclusion with a suitable reference relying on the relationship between drainage bubble speed and liquid viscosity. For instance, critical evaluation of role of viscosity and gas flowrate on slag foaming by Barella et al. or similar works.

Best Regards

We thank the reviewer for raising this point. Now we have added a suitable reference with the reviewer’s warm reminders. (Lines 395-402, changes highlighted in red)

Looking forward to your decision,

With kind regards,

Sincerely yours,

Mr. Hang Hu.

School of Minerals Processing and Bioengineering, Central South University, Changsha 410083.

E-mail address: [email protected]

Reviewer 3 Report

The changes that you have made have improved the quality of the presentation and the discussion. This paper can now be accepted for publication

Author Response

(The authors gave the same response as above.)
